

# Differential transcript profile of inhibitors with potential anti-venom role in the liver of juvenile and adult *Bothrops jararaca* snake

Cícera Maria Gomes[1,2], Karen de Morais-Zani[1,2], Stephen Lu[3],
Diego de Souza Buarque[3], Glória Regina Cardoso Braz[4,5],
Kathleen Fernandes Grego[1], Aparecida Sadae Tanaka[5,6,*] and
Anita Mitico Tanaka-Azevedo[1,2,*]

[1] Laboratório de Herpetologia, Instituto Butantan, São Paulo, Brazil
[2] Interunidades em Biotecnologia, Universidade de São Paulo, São Paulo, Brazil
[3] Departamento de Bioquímica, Universidade Federal de São Paulo, São Paulo, Brazil
[4] Departamento de Bioquímica, Universidade Federal do Rio de Janeiro, Rio de Janeiro, Brazil
[5] Instituto Nacional de Ciência e Tecnologia em Entomologia Molecular, São Paulo, Brazil
[6] Escola Paulista de Medicina / Departamento de Bioquímica, Universidade Federal de São Paulo, São Paulo, Brazil
[*] These authors contributed equally to this work.

Corresponding authors
Aparecida Sadae Tanaka,
tanaka.bioq@epm.br
Anita Mitico Tanaka-Azevedo,
anita.azevedo@butantan.gov.br

## ABSTRACT

**Background**. Snakes belonging to the *Bothrops* genus are vastly distributed in Central and South America and are responsible for most cases of reported snake bites in Latin America. The clinical manifestations of the envenomation caused by this genus are due to three major activities—proteolytic, hemorrhagic and coagulant—mediated by metalloproteinases, serine proteinases, phospholipases $A_2$ and other toxic compounds present in snake venom. Interestingly, it was observed that snakes are resistant to the toxic effects of its own and other snake's venoms. This natural immunity may occur due the absence of toxin target or the presence of molecules in the snake plasma able to neutralize such toxins.

**Methods**. In order to identify anti-venom molecules, we construct a cDNA library from the liver of *B. jararaca* snakes. Moreover, we analyzed the expression profile of four molecules—the already known anti-hemorrhagic factor Bj46a, one gamma-phospholipase $A_2$ inhibitor, one inter-alpha inhibitor and one C1 plasma protease inhibitor—in the liver of juvenile and adult snakes by qPCR.

**Results**. The results revealed a 30-fold increase of gamma-phospholipase $A_2$ inhibitor and a minor increase of the inter-alpha inhibitor (5-fold) and of the C1 inhibitor (3-fold) in adults. However, the Bj46a factor seems to be equally transcribed in adults and juveniles.

**Discussion**. The results suggest the up-regulation of different inhibitors observed in the adult snakes might be a physiological adaptation to the recurrent contact with their own and even other snake's venoms throughout its lifespan. This is the first comparative analysis of ontogenetic variation of expression profiles of plasmatic proteins with potential anti-venom activities of the venomous snake *B. jararaca*. Furthermore, the present data contributes to the understanding of the natural resistance described in these snakes.

## INTRODUCTION

The genus *Bothrops* is widely distributed in Central and South America, being the most common genus reported in ophidian accidents (*Cidade et al., 2006*). In Brazil, the species *Bothrops jararaca* (*B. jararaca*) accounts for the majority of the 30,000 cases of envenomation registered annually (*Ministério da Saúde, 2016*), due to its abundance and broad geographical distribution.

Clinical manifestations of bothropic envenomation are due to the following venom activities: (1) proteolytic, resulting in inflammatory edema at the bite site; (2) hemorrhagic, related to endothelial damage and systemic bleeding; (3) coagulant, responsible for the consumption of coagulation factors and consequent homeostasis disruption; and (4) myonecrotic, related to permanent tissue loss, disability and amputation (*Rosenfeld, 1971*). These activities are mediated by a number of venom components, such as metalloproteinases, serine proteinases, phospholipases A$_2$ (PLA$_2$s), L-amino acid oxidases (LAAOs) and other toxic compounds (*Fox et al., 2006*; *Zelanis et al., 2010*). The quantitative and qualitative composition of toxins present in snake venoms may vary according to several factors, such as ontogenetic development (*Zelanis et al., 2010*), seasonal period (*Williams & White, 1992*), gender (*Menezes et al., 2006*), diet (*Gibbs & Mackessy, 2009*) and geographical distribution (*Alape-Giron et al., 2008*).

Another intriguing feature of the physiology of snakes is the "natural immunity" towards the toxicity of their own venom and other snake venoms. This resistance may be a result of a mutation in the gene encoding the target of the venom toxin, rendering the target insensitive (*Burden, Hartzell & Yoshikami, 1975*; *Ohana et al., 1991*) and/or due to the presence of proteins that neutralize venom components in the blood of resistant animals (*Clark & Voris, 1969*; *Omori-Satoh, 1977*; *Omori-Satoh et al., 1972*; *Straight, Glenn & Snyder, 1976*). This inter- and intra-species resistibility makes snake plasma an interesting and rich source of bioactive compounds, since it can be explored for the isolation of proteins that can neutralize the toxic components of snake venoms and can contribute to the development of new approaches for the treatment of ophidic accidents (*De Morais-Zani et al., 2013*; *Lizano, Domont & Perales, 2003*).

It is believed that studies on the natural resistibility of snakes began with *Fontana (1781)* who stated that "the venom of the viper is not venomous to its species", more than 230 years ago. Eighty years after this pioneer report, *Guyon (1861)* discovered that the natural immunity is not species-specific. Since the observations made by *Fontana (1781)*, a number of "plasma factors" have been identified, isolated and characterized, not only from venomous and non-venomous snakes (*Thwin et al., 2000*) but also from different animals (*Fortes-Dias, 2002*; *Omori-Satoh, Yamakawa & Mebs, 2000*; *Thwin & Gopalakrishnakone, 1998*).

In this context, *Nahas et al. (1973)* were the first to identify the presence of a natural inhibitor in the plasma of *B. jararaca* in 1973. Later, *Nahas et al. (1983)* have also described the "inactivating effect" of *B. jararaca* plasma upon the coagulant activity of venom from 27 different snake species. Several inhibitors have already been identified in *B. jararaca* plasma and serum. The first molecule isolated from the plasma of this species, to our knowledge, was described by *Tanizaki et al. (1991)* and has the ability to inhibit the hemorrhagic and caseinolytic activity of *B. jararaca* whole venom. Further, this molecule was reported to also inhibit the venom pro-coagulant activity and lethality (*De Oliveira & Tanizaki, 1992*). Besides, an anti-hemorrhagic factor, Bj46a, a potent inhibitor of metalloproteinases and venom hemorrhagic activity, was also purified from *B. jararaca* serum (*Valente et al., 2001*). In addition, some PLA$_2$s inhibitors (PLIs) are identified in *B. jararaca* plasma through proteomic analysis (2D SDS-PAGE and mass spectrometry) (*De Morais-Zani et al., 2013*). Interestingly, a comparative study of the plasma composition of juvenile and adult *B. jararaca* snakes showed that the inhibitors aforementioned (Bj46a and PLIs) might be present at different levels during ontogenetic development and that this variability can be related to the ontogenetic shift described in its venom (*De Morais-Zani et al., 2013*).

Although there is an increasing interest in the natural resistance of snakes against venom toxins, the knowledge about snake plasma constitution is still sparse. Therefore, we constructed a liver cDNA library from *B. jararaca* snakes and compared the expression profile of possible anti-venom molecules between adults and juvenile snakes. The results described herein can open perspectives to the design of new molecules for therapeutic and biotechnological purposes and to the development of new strategies to the management of snake envenomation.

## METHODS

### Ethics statement

Experimental protocols using animals have been conducted in agreement with the Ethical Principles in Animal Research adopted by the Brazilian College of Animal Experimentation and were approved by the Ethical Committee for Animal Research of Butantan Institute (CEUAIB) under registry No. 794/11 and No. 931/12.

### *B. jararaca* liver collection

*B. jararaca* specimens were obtained from Herpetology Laboratory of Butantan Institute (São Paulo—Brazil). Eight females were used, five adults and three juveniles, all from São Paulo State, Brazil. Snakes were euthanized by intracoelomic administration of thiopental (90 mg kg$^{-1}$) and lidocaine hydrochloride (5 mg kg$^{-1}$). The livers were immediately dissected and stored in liquid nitrogen for cDNA library construction. For qPCR experiments, livers were stored in Trizol (Invitrogen, Carlsbad, CA, USA) and kept in −80 °C until use.

### cDNA library construction and sequencing

The mRNA was isolated from the liver of two adult *B. jararaca* snakes using the RNAeasy Mini Kit (Qiagen, Hilden, Germany). Thereafter the cDNA library was constructed

using the SMART cDNA Library Construction Kit (Clontech, Mountain View, CA, USA) according to manufacturer's instructions. The *E. coli* BM 25.8 strain was inoculated in 2 mL of LB medium and incubated at 31 °C until the $OD_{600}$ reached 1.3, followed by the addition of $MgCl_2$ (10 mM) and the amplified cDNA library. Infected bacterial cells were spread on LB plates containing ampicillin and growth overnight at 31 °C. Random isolate clones were selected and used for mini plasmid preparation, in which 200 ng of each plasmid were combined with 10 pmol of LD-5′ primer (5′–CTCGGGAAGCGCGCCATTGTGTTGGT –3′) and BigDye reagent (Applied Biosystems, Foster City, CA, USA) in a final volume of 10 µL. Reactions were submitted to 36 cycles of 96 °C –10 s, 50 °C –10 s and 60 °C –4 min, followed by precipitation with ethanol and sodium acetate buffer. DNA sequencing was carried on an ABI 3130 sequencer (Applied Biosystems, Foster City, CA, USA) as described by *Buarque et al. (2013)*.

## Bioinformatics analysis

Bioinformatics analysis was performed as previously described (*Karim, Singh & Ribeiro, 2011*). The software was written and provided by Dr. José Marcos Ribeiro (NIAID –NIH) in Visual Basic 6.0 (Microsoft). The functional annotation of CDS was performed through Blastn and Blastx (*Altschul et al., 1990*) against several databases (non-redundant protein, refseq-invertebrate, refseq-protozoa, refseq-vertebrate from NCBI and the custom made LEPIDOSAURIA database). The functionally annotated sequences were plotted in an excel spreadsheet Data S1.

## Quantitative PCR (qPCR)

Quantitative PCR was performed using three biological samples for each group (juveniles and adults). Total RNA was extracted from the liver of adults and juveniles *B. jararaca* snakes using Trizol (Invitrogen, Carlsbad, CA, USA) and quantified using NanoVue equipment (GE Healthcare). Total RNA was treated with one unit of DNase (Fermentas, Waltham, MA, USA) for 1 h at 37 ° C. Reactions were stopped by adding EDTA and heating for 10 min at 65 °C. cDNA synthesis was performed using the ImProm-II$^{TM}$ Reverse Transcription System (Promega, Madison, WI, USA) following the manufacturer's guidelines and qPCR was performed according *Livak & Schmittgen (2001)*, using the SYBR$^{®}$ Green PCR Master Mix (Applied Biosystems, Foster City, CA, USA) in a 7500 Real-Time PCR System (Applied Biosystems, Foster City, CA, USA). The qPCR reaction was performed using: 1 µL of 10X diluted cDNA, 6 µL of SYBR$^{®}$ Green and 150 nM of each specific-primers: Bj46a gene (Bj46a forward and Bj46a reverse), γ-PLI gene (γ-PLI forward and γ-PLI reverse), inter-alpha inhibitor (inter-alpha inhibitor forward and inter-alpha inhibitor reverse) and plasma protease C1 inhibitor-like (C1- forward and C1- reverse), in a 12 µL final volume. Primers sequences are listed in Data S2. β-actin gene was used as the internal control. The PCR program comprised 40 cycles at 94 °C (15 s) and 60 °C (1 min), followed by melt curve generation. Melt curves were analyzed to check the specificity of amplification. Reactions were performed in triplicate (for each biological sample) and all values are represented as the mean $\pm$ standard deviation. An unpaired $t$ test was conducted for statistical analysis, and a significant difference was accepted at $p < 0.05$.

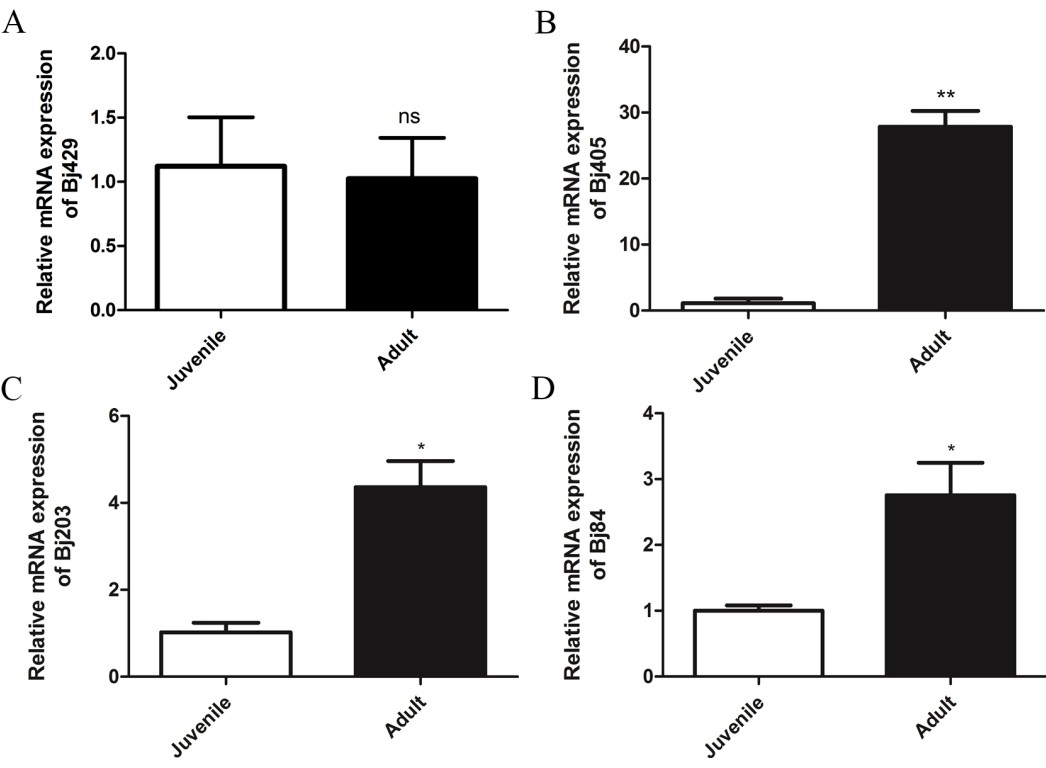

**Figure 1 Expression analysis of plasmatic inhibitors from juvenile and adult *B. jararaca* snakes.** The abundance expression of (A) BJ46a-like (Bj429), (B) γ- phospholipase $A_2$ inhibitor (Bj405), (C) inter-alpha-trypsin inhibitor (Bj203) and (D) plasma protease C1-inhibitor (Bj84). Error bars represent the standard deviation of the mean from three independent experiments ($n = 3$). Statistical analysis was carried with unpaired *t test*. Asterisks represent significant difference: $*p < 0.05$ and $**p < 0.01$. NS, non-statistical significant.

## RESULTS

### Anti-hemorrhagic factor BJ46a

Transcripts encondig to metalloprotease inhibitors were found in our cDNA library of *B. jararaca liver* Data S1, including the anti-hemorrhagic factor BJ46a, which presents inhibitory activity against venom metalloproteases. Quantitative analyses obtained by qPCR using specific primers for BJ46a showed no significant differences between juvenile and adult *B. jararaca* snakes (Fig. 1A). The amino acid sequence deduced from our cDNA library (amino acids residues 137–345) was aligned against similar proteins described in other snake species (Fig. 2) and reinforces the similarity among several related-molecules described in different Viperidae snakes. However, it is important to note that the partial sequence of this transcript contains specific amino acid residues related to BJ46a (an anti-hemorrhagic factor from *B. jararaca*) and HLP-B (a snake fetuin with no anti-hemorrhagic activity from *Gloydius brevicaudus*), as well as exclusive amino acid residues Data S3. Therefore, the molecule described herein is referred as BJ46a-like.

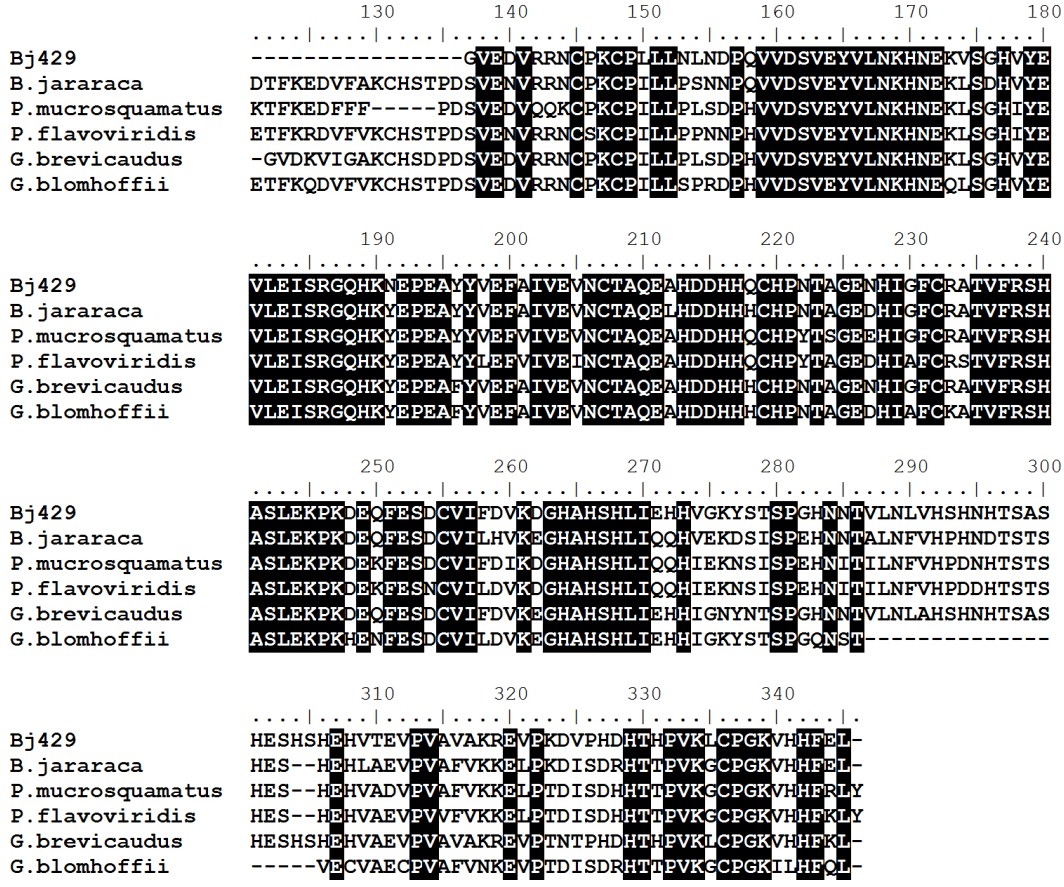

```
                          130       140       150       160       170       180
                  ....|....|....|....|....|....|....|....|....|....|....|....|
Bj429             ----------------GVEDVRRNCPKCPLLLNLNDPQVVDSVEYVLNKHNEKVSGHVYE
B.jararaca        DTFKEDVFAKCHSTPDSVENVRRNCPKCPILLPSNNPQVVDSVEYVLNKHNEKLSDHVYE
P.mucrosquamatus  KTFKEDFFF-----PDSVEDVQQKCPKCPILLPLSDPHVVDSVEYVLNKHNEKLSGHIYE
P.flavoviridis    ETFKRDVFVKCHSTPDSVENVRRNCSKCPILLPPNNPHVVDSVEYVLNKHNEKLSGHIYE
G.brevicaudus     -GVDKVIGAKCHSDPDSVEDVRRNCPKCPILLPLSDPHVVDSVEYVLNKHNEKLSGHVYE
G.blomhoffii      ETFKQDVFVKCHSTPDSVEDVRRNCPKCPILLSPRDPHVVDSVEYVLNKHNEQLSGHVYE

                          190       200       210       220       230       240
                  ....|....|....|....|....|....|....|....|....|....|....|....|
Bj429             VLEISRGQHKNEPEAYYVEFAIVEVNCTAQEAHDDHHQCHPNTAGENHIGFCRATVFRSH
B.jararaca        VLEISRGQHKYEPEAYYVEFAIVEVNCTAQELHDDHHHCHPNTAGEDHIGFCRATVFRSH
P.mucrosquamatus  VLEISRGQHKYEPEAYYVEFVIVEVNCTAQEAHDDHHQCHPYTSGEEHIGFCRATVFRSH
P.flavoviridis    VLEISRGQHKYEPEAYYLEFVIVEINCTAQEAHDDHHQCHPYTAGEDHIAFCRSTVFRSH
G.brevicaudus     VLEISRGQHKYEPEAFYVEFAIVEVNCTAQEAHDDHHHCHPNTAGENHIGFCRATVFRSH
G.blomhoffii      VLEISRGQHKYEPEAFYVEFAIVEVNCTAQEAHDDHHHCHPNTAGEDHIAFCKATVFRSH

                          250       260       270       280       290       300
                  ....|....|....|....|....|....|....|....|....|....|....|....|
Bj429             ASLEKPKDEQFESDCVIFDVKQGHAHSHLIEHHVGKYSTSPGHNNTVLNLVHSHNHTSAS
B.jararaca        ASLEKPKDEQFESDCVILHVKEGHAHSHLIQQHVEKDSISPEHNNTALNFVHPHNDTSTS
P.mucrosquamatus  ASLEKPKDEKFESDCVIFDIKQGHAHSHLIQQHIEKNSISPEHNITILNFVHPDNHTSTS
P.flavoviridis    ASLEKPKDEKFESNCVILDVKQGHAHSHLIQQHIEKNSISPEHNITILNFVHPDDHTSTS
G.brevicaudus     ASLEKPKDEQFESDCVIFDVKEGHAHSHLIEHHIGNYNTSPGHNNTVLNLAHSHNHTSAS
G.blomhoffii      ASLEKPKHENFESDCVILDVKEGHAHSHLIEHHIGKYSTSPGQNST-------------

                          310       320       330       340
                  ....|....|....|....|....|....|....|....|....|....|.
Bj429             HESHSHEHVTEVPVAVAKREVPKDVPHDHTHPVKLCPGKVHHFEL-
B.jararaca        HES--HEHLAEVPVAFVKKELPKDISDRHTTPVKGCPGKVHHFEL-
P.mucrosquamatus  HES--HEHVADVPVAFVKKELPTDISDHHTTPVKGCPGKVHHFRLY
P.flavoviridis    HES--HEHVAEVPVVFVKKELPTDISDHHTTPVKGCPGKVHHFKLY
G.brevicaudus     HESHSHEHVAEVPVAVAKREVPTNTPHDHTHPVKLCPGKVHHFKL-
G.blomhoffii      -----VECVAECPVAFVNKEVPTDISDRHTTPVKGCPGKILHFQL-
```

**Figure 2   Multiple alignments of amino acid sequences of Bj46a-like (Bj429) with similar sequences described in different species of snakes.** The sequences used are from *Bothrops jararaca* (sp|Q9DGI0.1), *Protobothrops mucrosquamatus* (XP_015681073.1), *Protobothrops flavoviridis* (sp|P29695.2), *Gloydius brevicaudus* (sp|Q5KQS2.1) and *Gloydius blomhoffii* (sp|Q5KQS1.1). Identical residues are black boxed.

## Gamma phospholipase A₂ inhibitor

Gamma-phospholipase A$_2$ inhibitor (γ-PLI) expression profile analysis by qPCR reveals an up-regulation around 30-fold in adults in relation to juvenile specimens (Fig. 1B). From our cDNA library, it was possible to deduce the whole inhibitor amino acid sequence (Fig. 3). When aligned to the sequence of a previously reported *B. jararaca* γ-PLI, the two sequences differ only by four amino acids residues in the positions 48 (G →A), 200 (F →I), 201 (K →R) and 203 (T →A). Note that the amino acid position numbers correspond to the alignment of several γ-PLI displayed in Fig. 3, which showed a high degree of similarity. It is interesting to observe the high incidence of amino acid substitutions found in the C-terminal region, not only between the two γ-PLI described in *B. jararaca*, but among the nine inhibitors aligned, described in three different genera of snakes from Viperidae (*Bothrops* and *Protobothrops*) and Colubridae families (*Elaphe*).

## Inter-alpha inhibitor

Transcripts related to the serine protease inhibitor inter-alpha inhibitor presented a 5-fold up-regulation in the liver of adult *B. jararaca* snakes (Fig. 1C). The partial amino acid

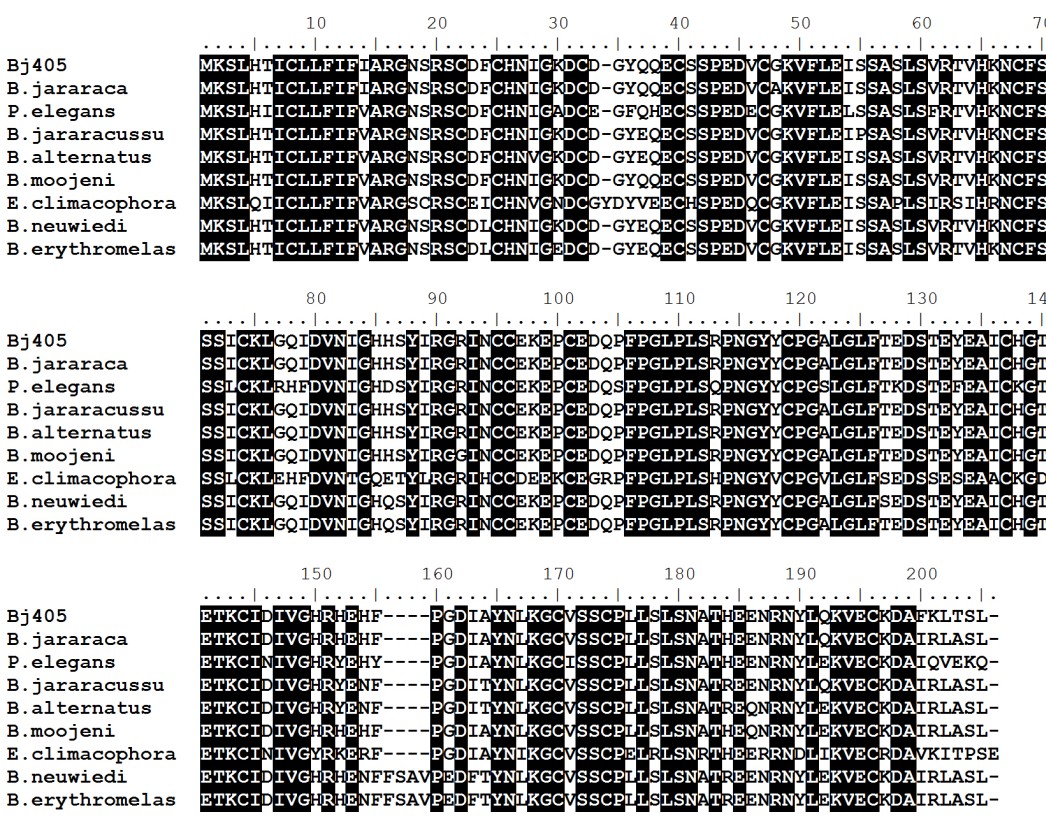

**Figure 3** Multiple alignments of amino acid sequences of γ- phospholipase A$_2$ inhibitor (Bj405) with similar sequences described in different species of snakes. The sequences used are from *Bothrops jararaca* (gb|ABV91331.1), *Protobothrops elegans* (dbj|BAJ14719.1), *Bothrops jararacussu* (gb|ABV91333.1), *Bothrops alternatus* (gb|ABV91326.1), *Bothrops moojeni* (gb|ABV91334.1), *Elaphe climacophora* (dbj|BAH47550.1), *Bothrops neuwiedi* (gb|ABV91336.1) and *Bothrops erythromelas* (gb|ABV91328.1). Identical residues are black boxed.

sequence of inter-alpha inhibitor heavy chain (H3-like) was deduced from a nucleotide sequence found in our cDNA library, and by our knowledge this is the first description of this molecule in *B. jararaca*. The inter-alpha inhibitor sequence showed similarity to the protein described in several reptile species, such as non-venomous and venomous snakes (*Python bivittatus* and *Protobothrops mucrosquamatus*, respectively), lizards (*Anolis carolinensis* and *Gekko japonicus*) and turtles (*Pelodiscus sinensis* and *Chrysemys picta bellii*) (Fig. 4).

## Plasma protease C1 inhibitor

Transcripts encoding to plasma protease C1 inhibitor showed a 3-fold increased expression in the liver of adult *B. jararaca* in comparison to juvenile individuals (Fig. 1D). This is the first report on the presence of transcripts related to C1-inhibitor in *B. jararaca* liver. The C1-inhibitor C-terminal deduced amino acid sequence showed some degree of similarity to the molecule described in the lizard *Anolis carolinensis*, the alligator *Alligator mississipiensis* and in three different species of snakes belonging to Pythonidae, Colubridae and Viperidae families (*Python bivittatus*, *Thamnophis sirtalis* and *Protobothrops mucrosquamatus*) (Fig. 5).

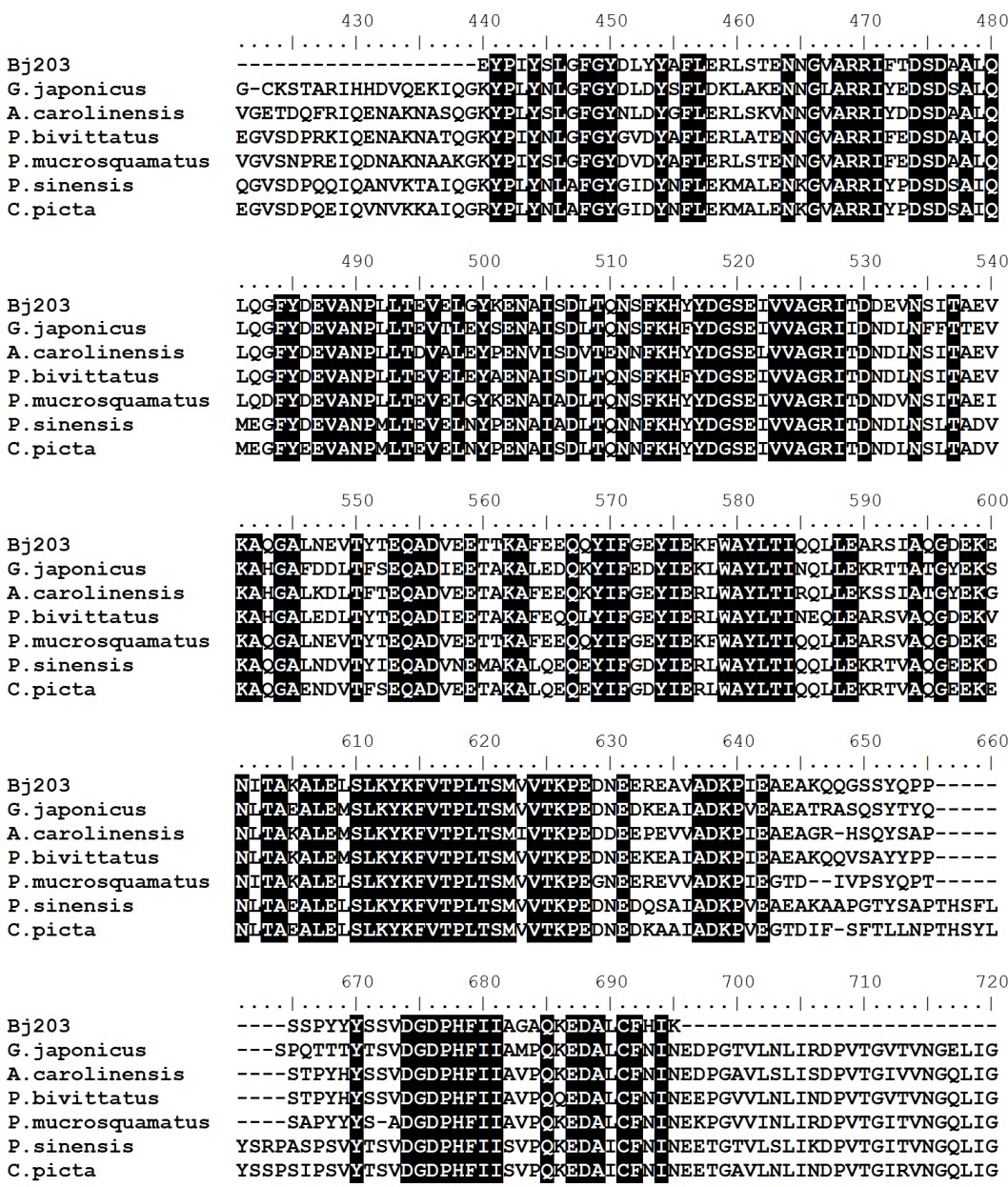

**Figure 4  Multiple alignments of amino acid sequences of inter-alpha-trypsin inhibitor (Bj203) with similar sequences described in different species of reptiles.** The sequences used are from *Gekko japonicus* (XP_015262960.1), *Anolis carolinensis* (XP_003217700.2), *Python bivittatus* (XP_007442992.1), *Protobothrops mucrosquamatus* (XP_015671353.1), *Pelodiscus sinensis* (XP_006127649.1) and from *Chrysemys picta bellii* (XP_008177427.1). Identical residues are black boxed.

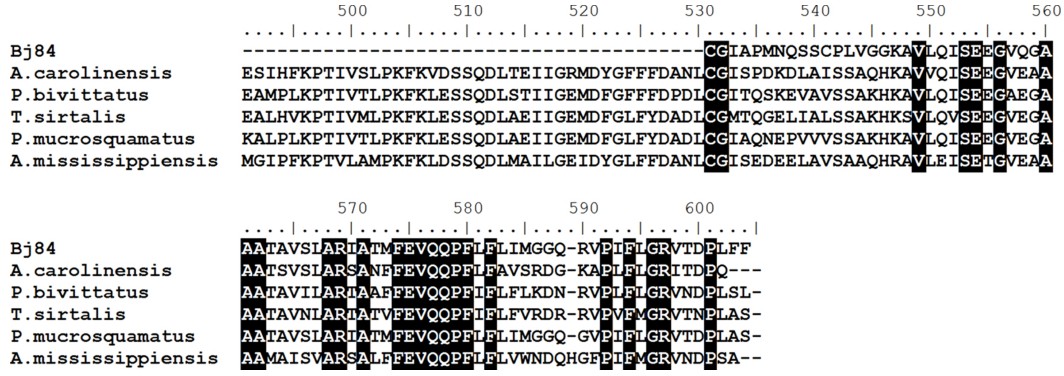

**Figure 5** **Multiple alignments of amino acid sequences of plasma protease C1 inhibitor (Bj84) with similar sequences described in different species of reptiles.** The sequences used are from *Anolis carolinensis* (XP_008109235.1), *Python bivittatus* (XP_007423129.1), *Thamnophis sirtalis* (XP_013930568.1), *Protobothrops mucrosquamatus* (XP_015676034.1) and *Alligator mississippiensis* (gb|KYO40723.1). Identical residues are black boxed.

When these sequences were aligned, the high variability in amino acid composition in the C-terminal region of C1-inhibitor among the species above mentioned is remarkable, as shown in Fig. 5.

## DISCUSSION

Although a number of snake venom gland transcriptomes have been characterized and are accessible in databases (for review, see *Brahma et al., 2015*) studies concerning gene expression in other tissues are scarce and only recently became available (*Castoe et al., 2011*; *Schwartz et al., 2010*). However, none of these studies focused on the quantitative analysis of inhibitors that might be involved on venom neutralization, with comparison of adult and juvenile profile. This comparative analysis may contribute to the elucidation of the physiology and anti-venom mechanisms described in *B. jararaca* plasma.

Snake venom metalloproteinases (SVMPs) are the most abundant components in adult and juvenile *B. jararaca* venom proteome and venom gland transcriptome (*Zelanis et al., 2012*; *Zelanis et al., 2016*), which displays hemorrhagic activity, as described for jararhagin (*Paine et al., 1992*), HF3 (*Assakura, Reichl & Mandelbaum, 1986*), bothropasin (*Queiroz et al., 1985*) and jararafibrase (*Maruyama et al., 1993*). Thus, the presence of inhibitory components in snake plasma may take part in the "accidental envenomation". This is the case of the anti-hemorrhagic factor BJ46a, a glycoprotein isolated from *B. jararaca* plasma that inhibits the hemorrhagic activity of its own venom, as well as the activity of isolated metalloproteinases jararhagin and atrolysin C (*Valente et al., 2001*).

A previous study evaluating the ontogenetic changes in the plasma proteomic profile of *B. jararaca* snakes showed that BJ46a is present in a higher relative abundance in the plasma of adult specimens (*De Morais-Zani et al., 2013*) suggesting a positive association with the higher hemorrhagic activity described in the venom of adult snakes (*Antunes et al., 2010*). However, the results presented herein showed no significant differences in BJ46a-like transcript levels between juvenile and adult *B. jararaca* snakes.

A concern that must be pointed out is the high similarity between anti-hemorrhagic factors (HSF from *Protobothrops flavoviridis* and BJ46a included) and other molecules with no inhibitory activity, i.e., HLP from *Protobothrops flavoviridis* and HLP-A and HLP-B from *Gloydius brevicaudus* (*Aoki et al., 2009*; *Yamakawa & Omori-Satoh, 1992*). These molecules are classified based on their primary structures as members of the fetuin family (*Aoki et al., 2009*).

The BJ46a-like transcript analyzed herein shows a high degree of similarity to BJ46a and HLP-B. In fact, the partial deduced sequence of this transcript contains specific amino acid residues related to BJ46a and HLP-B, as well as exclusive substitutions (Data S3). These findings indicate that in the present work we described a new molecule, although it is not possible to ensure its anti-hemorrhagic activity based only on its primary (and partial) amino acid sequence.

*Aoki et al. (2009)* concluded that HLP-B from *Gloydius brevicaudus* corresponds to the ortholog of the mammalian fetuins and that the anti-hemorrhagic factors HSF and HLP-A possibly evolved from this molecule in order to display a protective role against its own venom. In this context, it is possible that the new molecule describe in this work represents this transition in *B. jararaca*. Nevertheless, the mechanisms underlying the regulation of the expression of BJ46a and BJ46a-like molecules in juvenile and adult *B. jararaca* snakes need to be elucidated.

Three structural classes of PLIs have been described in snake plasma: (1) $\alpha$-PLIs, which inhibit specifically acidic PLA$_2$s from group II (found in the venom of Viperidae snakes), (2) $\beta$-PLIs, which inhibit specifically basic PLA$_2$s from group II, and (3) $\gamma$-PLIs, which show inhibitory activity towards group I (from venom of Elapidae, Hydrophiidae and Colubridae snakes) and II PLA$_2$s (*Estevao-Costa et al., 2008*; *Inoue et al., 1997*; *Kinkawa et al., 2010*). Considering the broad spectrum of pharmacological activities displayed by snake venom PLA$_2$s, as neurotoxicity, myotoxicity, edema-inducing and anticoagulant activities, the presence of PLIs in the plasma of these animals is of paramount importance. In addition to the key role of these PLIs for snake physiology, the identification and characterization of PLIs are of great interest for biotechnological purposes, especially those called MIPs (myotoxin inhibitor toxin), which target specifically basic Asp49- and Lys49-PLA$_2$s, responsible for local myonecrosis, one of the most serious consequences of *B. jararaca* envenoming (*Campos et al., 2016*; *Mora-Obando et al., 2014*; *Santoro et al., 2008*).

In the present work, we analyzed the transcriptional profile of a $\gamma$-PLI in the liver of juvenile and adult *B. jararaca* snakes. qPCR results showed that the levels of transcripts encoding to $\gamma$-PLI was 30-fold higher in adult than those observed in juvenile specimens. At a first glance, these are unexpected results, since a previous plasma proteomic analysis indicated that $\gamma$-PLI are found in a higher relative abundance in the plasma of juvenile *B. jararaca* (*De Morais-Zani et al., 2013*) and the venom of juvenile specimens also displayed higher catalytic PLA$_2$ activity (*Antunes et al., 2010*). However, a study conducted by *Kinkawa et al. (2010)* showed that the gene expression of $\alpha$-, $\beta$- and $\gamma$-PLIs was induced by the intramuscular injection of venom in the venomous snake *Gloydius brevicaudus*. Therefore, the higher expression levels of $\gamma$-PLI in adult *B. jararaca* liver described herein might be the result of the physiological response of the snakes to the repeatedly contact with

their own venom during their development. However, we can not discard the possibility that this increase is under genetic control, a hypothesis that will be discussed further in this article.

The previously reported *B. jararaca* γ-PLI (*Estevao-Costa et al., 2008*) and the γ-PLI reported herein differ only in four of 200 amino acids (including the 19-residue signal peptide), showing that we described for the first time a new isoform of this inhibitor.

The presence of γ-PLI isoforms was described in *Lachesis muta* (*Fortes-Dias, Barcellos & Estevao-Costa, 2003*) and in several *Bothrops* species (*Estevao-Costa et al., 2008*). The variability in amino acid sequence and in inhibitory activity of these molecules is not surprising, considering that snake venoms that contains PLA$_2$ are often composed by a mixture of isoforms of these enzymes (*Estevao-Costa et al., 2008*). Therefore, the presence of γ-PLI isoforms in snake plasma may be of great physiological importance, ensuring the efficiency of these molecules and broadening their range of inhibition (*Estevao-Costa et al., 2008*).

Besides a number of amino acid substitutions were present amongst γ-PLI described in species from *Bothrops* genera, a highly conserved region composed by 15 amino acid residues ($^{104}$QPFPGLPLSRPNGYY$^{118}$) is present in all of them and it was recognized as the consensus motif most probably involved in the interaction of γ-PLI with snake venom PLA$_2$ (*Estevao-Costa et al., 2008*).

This 15-residue sequence, as well as the 16 cysteines in the mature protein and the proline-rich region (residues 100–120 in Fig. 3), which play an important role in maintaining the conformation of the interaction sites of the molecule (*Dunn & Broady, 2001*; *Kini, 1998*), are conserved in the molecule described herein, reinforcing its PLA$_2$ inhibitory activity and protective role in snake plasma. The four amino acids substitutions observed in the molecule described in the present work in the positions 48 (A →G), 200 (I →F), 201 (R →K) and 203 (A →T) (the last two also described in γ-PLI of *E. climacophora*), may exert little or no influence in its inhibitory activity, since these residues are not identified (to date) as key residues for the biological activity of the molecule.

Snake venom serine proteinases (SVSPs) are another important group of toxins that play a central role in the envenomation caused by *B. jararaca* snake. These enzymes affect mainly the hemostatic system, acting on the components of the coagulation cascade and on the fibrinolytic and kallilrein-kinin systems (*Serrano, 2013*). In terms of relative abundance, SVSPs occupy the second position in the venom proteome of this species (*Zelanis et al., 2016*). Due to the central activity displayed by these toxins, we selected two serine proteinase inhibitors to evaluate the level of their related transcripts in juvenile and adult *B. jararaca* snakes.

Inter-alpha inhibitors constitute a family of proteins that acts in the regulation of the inflammatory process and plays a role in wound healing (*Kobayashi, 2006*; *Lim, 2013*). These molecules broadly inhibit serine proteases, decrease pro-inflammatory and enhance anti-inflammatory mediators and block complement activation during systemic inflammation (*Fries & Kaczmarczyk, 2003*; *Okroj et al., 2012*). These inhibitors can be found in plasma as inter alpha inhibitor, which is composed by two heavy chains (H1 and H2) and one light chain (LC), or as pre-alpha inhibitor, which consists of one heavy (H3) and one light chain (LC) (*Fries & Blom, 2000*). In this work, we found that transcripts related to the heavy chain of pre-alpha inhibitor (H3-like) presented a 5-fold up-regulation
in the liver of adult *B. jararaca* snakes. It seems plausible that this inhibitor can play a role on the neutralization of the major and minor activities of SVSP, such as disturbance of hemostasis and induction of inflammatory reactions, respectively.

Another important plasma serine proteinase inhibitor is C1 inhibitor, a multi-functional molecule that acts inactivating a number of serine proteases in different enzymatic cascades, as complement, coagulation, and fibrinolytic systems (*Ghannam et al., 2016*). It was hypothesized that this inhibitor could be involved in the neutralization of venom components in case of accidental envenomation. In addition to the impact on blood coagulation, *B. jararaca* SVSP can activate the complement system and, consequently, generate anaphylatoxins that might play a key role in the inflammatory process and also contribute to the spreading of other venom toxins (*Pidde-Queiroz et al., 2010*). Therefore, we decided to evaluate the levels of transcripts related to C1 inhibitor in the liver of *B. jararaca* snakes. Results described herein showed that transcripts encoding to this plasma inhibitor showed a 3-fold increase in the liver of adult specimens in comparison to juvenile individuals.

The analysis of ontogenetic variation in venom activities of *B. jararaca* showed that the activity of serine proteinases is slightly higher in adult individuals (*Antunes et al., 2010*), which could justify the higher expression of serine proteinase inhibitors, as inter-alpha inhibitor and C1 inhibitor, found in the liver of adult snakes.

Besides the possible role in neutralization of venom components, plasmatic serine proteinase inhibitors play a fundamental role in hemostasis maintenance in reptiles. Reptiles have armored skin, low blood pressure, slow circulation time, and are commonly lethargic in their movements. Therefore, their arterial blood moves in conditions more like those of the mammal venous side, which would favor the occurrence of intravascular clotting (*Tanaka-Azevedo, Tanaka & Sano-Martins, 2003*). Nevertheless, blood coagulation in reptiles is slower than in mammals, due to the absence or low concentration of some coagulation factors (such as VIII and IX) (*Didisheim, Hattori & Lewis, 1959*) and the presence of specific inhibitors (*Morais et al., 2009*; *Tanaka-Azevedo, Tanaka & Sano-Martins, 2003*; *Tanaka-Azevedo et al., 2004*).

However, it is noteworthy that 3 out of 4 mRNA related to the plasmatic proteins studied in the present work are more expressed in the liver of adult snakes. Bearing in mind the results described by *Kinkawa et al. (2010)*, regarding the control of PLIs expression, and taken together the results presented in herein, it is tempting to suggest that the higher expression levels of γ-PLI, inter-alpha inhibitor and C1-inhibitor observed in adult snakes might be a natural physiological response of the snakes to the recurrent contact with their own venom throughout the life.

Nevertheless, venom neutralizing factors are identified in the serum of neonates of the ophiophagus snake *Clelia clelia*, before any contact with venom during feeding on venomous snakes. This observation shows that, at least for this species, the contact with venoms is not the driving factor for the expression of plasmatic molecules with antivenom properties, being under genetic control. In this context, another possibility to be considered is that the expression of transcripts related to protease inhibitors in *B. jararaca* is under a genetically programmed control, being predetermined to be highly expressed in the adult stage. Indeed, bites between snakes are more prone to happen among adult individuals,

especially during the period of copulation, when the encounter and contact between snakes are more frequent (SS Sant'Anna, pers. comm., 2017). Nevertheless, these conclusions are speculative and it is important to emphasize that complementary studies are necessary to support these hypotheses.

In summary, this work provides the first comparative analysis of ontogenetic variation of expression profiles of plasmatic proteins with potential anti-venom activities of the venomous snake *B. jararaca*. Our data contributes to the understanding of the natural resistance against "self-envenomation" described in these snakes and provide new target molecules with biotechnological potential that can be useful for the development of new approaches for the treatment of ophidic accidents.

## ACKNOWLEDGEMENTS

We are grateful to Ms. Jacilene Barbosa da Silva Monteiro for technical assistance in DNA sequencing and Real Time PCR experiments (INFAR, Escola Paulista de Medicina, Universidade Federal de São Paulo, Brazil).

### Funding

This work was supported by the Fundação de Amparo à Pesquisa do Estado de São Paulo (FAPESP) (Proc. No 2010/10266-5, 2012/03657-8, 2013/05357-4 and 2014/11108-0), the Conselho Nacional de Desenvolvimento Tecnológico (CNPq) (Proc. No. 308780/2013-2) and the INCT—Entomologia Molecular. The funders had no role in study design, data collection and analysis, decision to publish, or preparation of the manuscript.

### Grant Disclosures

The following grant information was disclosed by the authors:
Fundação de Amparo à Pesquisa do Estado de São Paulo (FAPESP): 2010/10266-5, 2012/03657-8, 2013/05357-4, 2014/11108-0.
Conselho Nacional de Desenvolvimento Tecnológico (CNPq): 308780/2013-2.
INCT—Entomologia Molecular.

### Competing Interests

The authors declare there are no competing interests.

### Author Contributions

- Cícera Maria Gomes performed the experiments, analyzed the data, wrote the paper.
- Karen de Morais-Zani wrote the paper, reviewed drafts of the paper.
- Stephen Lu analyzed the data, wrote the paper, prepared figures and/or tables, reviewed drafts of the paper.
- Diego de Souza Buarque performed the experiments.
- Glória Regina Cardoso Braz analyzed the data.
- Kathleen Fernandes Grego conceived and designed the experiments.
- Aparecida Sadae Tanaka conceived and designed the experiments, analyzed the data, contributed reagents/materials/analysis tools, wrote the paper, prepared figures and/or tables, reviewed drafts of the paper.
- Anita Mitico Tanaka-Azevedo conceived and designed the experiments, contributed reagents/materials/analysis tools, wrote the paper, reviewed drafts of the paper.

**Animal Ethics**

The following information was supplied relating to ethical approvals (i.e., approving body and any reference numbers):

Experimental protocols using animals have been conducted in agreement with the Ethical Principles in Animal Research adopted by the Brazilian College of Animal Experimentation and were approved by the Ethical Committee for Animal Research of Butantan Institute (CEUAIB) under registry No. 794/11 and No. 931/12.

**Data Availability**

The raw data has been supplied as a Supplementary File.

**Supplemental Information**

Supplemental information for this article can be found online at http://dx.doi.org/10.7717/peerj.3203#supplemental-information.

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
