# Peer review of "Differential transcript profile of inhibitors with potential anti-venom role in the liver of juvenile and adult Bothrops jararaca snake"

_PeerJ, doi:10.7717/peerj.3203_

## Round 0.1 · original submission · Major Revisions

Reviewers have found your report to be original and of interest. However, as you will see in their comments, some issues need to be addressed. Please provide a point-by-point response on these comments and indicate all changes introduced in a revised version. As pointed out by reviewer #3, completion of the sequences for the inhibitors described would be highly desirable (and I would strongly encourage it) although I would not see this as an sine qua non requirement, considering the possibility of follow up studies on this subject. Also importantly, please address the observation from reviewer #2 when assuring that Bj429 corresponds to Bj46a, due to its high similarity to fetuin. Few grammatical errors/mispellings need to be revised.

Additional editorial comments

In both the Abstract and Introduction, the authors missed the relevant pathological effect of skeletal muscle necrosis (common in envenomings by Bothrops) which is mediated mainly by basic phospholipase A2 myotoxins. Please amend this omission.

Line 28: we constructed (instead of "we construct"; also to be corrected later in the body of the manuscript)

Line 35 (and in Conclusions): recurrent contact with own or else's snake venoms is one feasible explanation for the main observations herein reported - but are there other possibilities that remain open? Ontogenetic changes may not necessarily require an induction by contact with venom, and may simply be genetically "programmed". For example, in one study on antitoxic proteins present in the serum of the colubrid Clelia clelia, it was shown that a potent anti-hemorragic factor is present in neonate speciments (captive born) before any possible contact with venom: Lomonte et al. (1989) El suero de neonatos de Clelia clelia neutraliza la acción hemorrágica del veneno de Bothrops asper (Serpentes: Viperidae). Revista de Biología Tropical 38, 325-326.

Reviewer 1 ·

Basic reporting

In this work, Gomes et al. provide comparative analysis of ontogenetic variation of expression profiles of plasmatic proteins with potential anti-venom activities of Bothrops jararaca.

The manuscript is clear and well written, with sufficient introduction and background. All figures are relevant, high quality and well labelled.

There are some minor grammatical English errors.

Experimental design

This is and original study, with a relevant and well defined question. The ethical information is mentioned, and the methods are sufficiently detailed.

My only question is why does the methods mention that 5 adult snakes were used, but the results are based in only 3 of them?

Validity of the findings

Results are robust and the conclusion is well stated.

Reviewer 2 ·

Basic reporting

No Comments.

Experimental design

Methods for cDNA sequencing are not described in the text (Line 114-117).

Validity of the findings

Authors consider that Bj429 is antihemorrhagic factor Bj46a. But Bj429 is probably a snake fetuin with no inhibitory activity toward snake venom metalloproteases. Amino acid sequence of Bj429 is very similar to that of Gloydius brevicaudus HSF-Like protein-B (Q5KQS2.1 in Figure 2), which has been identified as a snake fetuin (Aoki, et. al, Toxicon 54 (2009) 481-490). Please read Aoki's paper. Snake sera contain some fetuin family proteins with different activities.
Authors are expected to find an alternative fetuin family protein Bj49a with antihemorrhagic activity and analyze its expression profile.

Additional comments

Ontogenetic expression profiles of anti-venom proteins are of particular interest and importance to understand the natural immunity of venomous snakes. This paper is a first report of ontogenetic expression of these proteins.

Reviewer 3 ·

Basic reporting

The authors report differential transcript profile of inhibitors with potential anti-venom role in the liver of juvenile and adult Bothrops jararaca snake. The authors constructed a cDNA library from the liver of B. jararaca snake to identified anti-venom molecules. The subject of this work is interesting but the proposed results are not complete and not convincing and the work requires additional experiments to justify the conclusions. The manuscript requires clarification and corrections. A native English speaker should improve the quality of writing.

Experimental design

• Why is only the partial Bj46a sequence identified in the cDNA library? The complete sequence should bedetermined. The sequence identity with protein Bj46a (previously identified in B jararaca) should be demonstrated.
• The authors identified a new sequence for the PLI-gamma inhibitor, which differs by 4 amino acid residues from the previously identified molecule. It is new and interesting data and the sequence of this new isoform should be deposited in the data bank.
What is the significance of the amino acid substitutions in this PLI isoform? Are they important for anti-PLA2 function or for formation of mature non-covalent complexes of PLI? Functional characterization of this PLI should be discussed.
• Only partial amino acid sequence of inter-alpha inhibitor heavy chain was determined in the cDNA library, why? If this molecule was identified for the first time in B. jararaca, the complete sequence should be presented and the function should be investigated.
• The authors reported a partial sequence of the plasma protease C1 inhibitor in the cDNA library. Once again, if this molecule was identified for the first time in B. jararaca, the complete sequence should be presented and the function should be investigated.

Validity of the findings

In conclusion, the results are highly speculative.

Additional comments

The authors should more completely identify the molecules before final interpretation of data.

---

## Round 0.2 · Minor Revisions

Most of the issues have been satisfactorily addressed. However, one of the reviewers (#2) has requested that you also address a minor point, please see his/her specific comment below.

Reviewer 2 ·

Basic reporting

No comment.

Experimental design

Although the qPCR reaction was also performed using PLI-alpha primers (Line 150), no comment nor result about PLI-alpha expression were given in this manuscript.

Validity of the findings

No comment.

---

## Round 0.3 · accepted · Accept

The explanation provided clarifies the specific query raised by the reviewer.